# Solvent engineering for scalable fabrication of perovskite/silicon tandem solar cells in air

Xuntian Zheng[1,4], Wenchi Kong[1,4] ✉, Jin Wen[1,4], Jiajia Hong[1], Haowen Luo[1], Rui Xia[2], Zilong Huang[1], Xin Luo[1], Zhou Liu[1], Hongjiang Li[2], Hongfei Sun[1], Yurui Wang[1], Chenshuaiyu Liu[1], Pu Wu[1], Han Gao[1], Manya Li[1], Anh Dinh Bui[3], Yi Mo[2], Xueling Zhang[2], Guangtao Yang[2], Yifeng Chen[2], Zhiqiang Feng[2], Hieu T. Nguyen ®[3], Renxing Lin[1], Ludong Li ®[1], Jifan Gao ®[2] ✉ & Hairen Tan ®[1] ✉

Perovskite/silicon tandem solar cells hold great promise for realizing high power conversion efficiency at low cost. However, achieving scalable fabrication of wide-bandgap perovskite (~1.68 eV) in air, without the protective environment of an inert atmosphere, remains challenging due to moisture-induced degradation of perovskite films. Herein, this study reveals that the extent of moisture interference is significantly influenced by the properties of solvent. We further demonstrate that n-Butanol (nBA), with its low polarity and moderate volatilization rate, not only mitigates the detrimental effects of moisture in air during scalable fabrication but also enhances the uniformity of perovskite films. This approach enables us to achieve an impressive efficiency of 29.4% (certified 28.7%) for double-sided textured perovskite/silicon tandem cells featuring large-size pyramids (2–3 μm) and 26.3% over an aperture area of 16 cm². This advance provides a route for large-scale production of perovskite/ silicon tandem solar cells, marking a significant stride toward their commercial viability.

Perovskite/silicon tandem solar cells stand out in the photovoltaics field due to their impressive power conversion efficiency (PCE) of 33.9% and cost-effectiveness, promoting them as prime candidates for rapid industrialization[1]. Among them, most of the high-efficiency perovskite/silicon tandem solar cells were fabricated on silicon heterojunction (SHJ) with mildly textured surface (pyramid height <1 μm), which enables the solution processing of wide-bandgap perovskite films on top[2–4]. Tandem solar cells utilizing double-side industrially textured SHJ bottom cells, featuring larger-sized pyramids (typically 2–3 μm), benefit from increased light capture[5,6]. However, the conventional one-step solution method for depositing perovskite films struggles to achieve full coverage over such pyramid tips due to the limited thickness (less than 1 μm), which is essential

for effective carrier transport, thus raising the risk of current shunts[7,8].

To address this issue, methods such as the full evaporation method (co-evaporation or sequential evaporation) and a hybrid two-step deposition method (combining evaporation of inorganic framework and solution processing of organic salts) have been developed. These methods enable the conformal deposition of perovskite films on industrially textured silicon solar cells[8–10]. Despite these advancements, the full evaporation method faces difficulties in incorporating additive engineering strategies, vital for enhancing the performance of perovskite solar cells, which results in a relatively lower performance compared with the hybrid two-step deposition[11]. Up to now, the record efficiency of perovskite/silicon tandem solar cells fabricated via the

[1]National Laboratory of Solid State Microstructures, College of Engineering and Applied Sciences, Frontiers Science Center for Critical Earth Material Cycling, Nanjing University, Nanjing 210023, China. [2]State Key Laboratory of PV Science and Technology, Trina Solar, ChangZhou 210031, China. [3]Research School of Electrical, Energy and Materials Engineering, College of Engineering and Computer Science, The Australian National University, Canberra, NSW, Australia. [4]These authors contributed equally: Xuntian Zheng, Wenchi Kong, Jin Wen. ✉e-mail: kongwenchi@nju.edu.cn; jifan.gao@trinasolar.com; hairentan@nju.edu.cn

hybrid method stands at 31.25%[12]. However, this impressive result has limited on lab-scale (~1 cm²) devices, fabricated under N₂ environment using spin-coating[7,8,12–17], which is not conducive to scalable production[18,19].

Given the commercial potential of perovskite/silicon tandem photovoltaics, it is critical to explore scalable fabrication methods that can be employed in ambient conditions. This necessitates a nuanced understanding of the influence of air moisture on perovskite crystallization, which plays a pivotal role in the film's quality[20–29]. Previous reports have suggested that moisture (RH = 30–40%) can promote the perovskite crystallization during annealing[28,30–36]. However, the strong hygroscopic properties of conventional isopropyl alcohol organic salt solution[37,38] will significantly compromise the quality of perovskite films in ambient air using a two-step method (Supplementary Fig. 1). Some strategies have been implemented to mitigate the impact of moisture on the fabrication of perovskite films, including the thermal-assisted methods like hot-cast methods and thermal radiation approaches[39,40]. However, these methods introduce heat treatment, resulting in additional energy consumption.

Here, we propose a solvent engineering strategy that leverages n-Butanol (nBA), characterized by its low polarity and saturation vapor pressure, as a solvent for organic salts. This approach effectively diminishes the detrimental effects of moisture in air on the fabrication of perovskite and improves their quality and uniformity on a large-scale silicon substrate. As a result, a single-junction solar cell with a wide bandgap of 1.68 eV achieves a PCE of 20.8% (0.049 cm²) and 19.6% (1.044 cm²). Furthermore, double-side textured perovskite/silicon tandem solar cells achieve an impressive efficiency of 29.4% for 1.044 cm² (certified 28.7%) and 26.3% for an aperture area of 16 cm². The encapsulated device retained 96.8% of the initial output after 780 h of maximum power point tracking. Additionally, we have showcased the potential for commercial scaling by achieving a conversion efficiency of 25.9% for 16 cm² devices fabricated via slot-die coating. This solvent engineering strategy demonstrates the feasibility of commercial perovskite/silicon tandem solar cells.

## Results

### Distinction of different alcohols as solvents

The perovskite films were fabricated by a two-step sequential deposition method based on previous work[15,23]. As depicted in Fig. 1a, our process combines co-evaporation and blade-coating techniques to meet the requirements for large-area fabrication of the perovskite films. Supplementary Fig. 2 shows the deposition of an inorganic framework on both glass and textured silicon substrates. It is worth noting that the second step was implemented in air to match the realistic production environment. However, ethanol and isopropyl alcohol, which are widely used as solvents of the organic salt in the second step, confront two major challenges in the natural environment: firstly, these solvents readily absorb environmental moisture[38]; secondly, the rapid evaporation rate of the solution will affect the film uniformity. Consequently, these challenges often result in inhomogeneous and poor perovskite films, adversely affecting the PCE and stability of the devices.

To address this issue, we carried out analysis and study on various alcohols with different saturated vapor pressures and polarities including ethyl alcohol (EA), isopropanol (IPA), n-butanol (nBA) and n-pentanol (nPA). The images of different solutions after adding organic salts to the alcohol are shown in Supplementary Fig. 3. For ease of expression, we refer to the following solutions, films and devices fabricated with ethanol as EA solution, EA film and EA device, as well as for IPA, nBA and nPA. As the carbon chain is lengthened, both the polarity of alcohols and the saturated vapor pressure decrease, as illustrated in Fig. 1b[41]. The saturated vapor pressure reflects the evaporation speed of the solvent, while the dielectric constant is positively related to the polarity of the solvent. Following

the principle that like dissolves like[42], the mutual solubility of alcohols and water—and thus their capacity to absorb moisture—is dictated by their polarity difference. Given water's high polarity, alcohols with greater polarity are more soluble in water, leading to increased water absorption.

To investigate the impact of moisture on these different alcohol solutions in air, we exposed a measured amount of each solution to open air and observed the changes. In air environment, moisture absorption leads to rapid oxidation of I⁻ to I₂, manifesting as a yellowing of the solution[43,44]. As shown in Fig. 1c, the EA and IPA solutions turned from colorless to light yellow after one hour of exposure, while nBA and nPA solutions exhibited no significant color change, underscoring the protective effect of low polarity solvents against moisture interference. Furthermore, we compared the films after blade-coating without gas quenching and annealing on the same substrate (glass/inorganic frame) and documented the changes photographically. Figure 1d illustrates that EA and IPA volatilize fastly and completely after blade-coating, in contrast to nBA and nPA films, which shows a gradual darkening. This shift signifies a decrease in volatilization rate with increasing carbon chain length, affecting perovskite crystallization dynamics. However, the slower volatilization rates results in the lingering of residual organic salts, which continues to undergo dissolution-recrystallization reactions with the perovskite[45]. This leads to localized accumulations of organic salts, as evidenced in Supplementary Figs. 4 and 5.

### Characterization of perovskite films fabricated by different alcohols

Supplementary Fig. 6 displays images of perovskite films fabricated using different alcohols, both in N₂ and air environments. These images corroborate the notion that moisture positively affects the crystallization rate of perovskite films[46], as inferred from the observable color changes. To further evaluate the effect of the volatilization rate of the solutions on the perovskite films formation, we compared the morphology and the structure of perovskite films by using a scanning electron microscope (SEM) and X-ray diffraction (XRD). Supplementary Fig. 7 reveals a pronounced PbI₂ signal in EA films before annealing, leading to a substantial amount of PbI₂ at the bottom of the perovskite layer (Fig. 2a and e). This indicates that the conversion from the inorganic framework to perovskite is incomplete. Such findings suggest that the delay of solvent volatilization rate is conducive to prolonging the reaction of inorganic frameworks with organic salt solutions in terms of promoting the transformation of the inorganic framework into perovskite.

Comparatively, perovskite films fabricated in air environment exhibit a heightened PbI₂ signal (Supplementary Fig. 8 and Fig. 2e), demonstrating that the moisture absorbed during fabrication prompts the decomposition of perovskite films upon air annealing. Specifically, the IPA films show a strong PbI₂ diffraction peak located at 12.6° (Fig. 2e), which stemmed from the decomposition of perovskite films after air annealing at 35% relative humidity—a finding consistent with SEM image of the IPA films (Fig. 2b). Impressively, the nBA films exhibited the lowest intensity of PbI₂ peak in Fig. 2e, with minimal residual PbI₂ particles observed on the surface (Fig. 2c), indicating negligible perovskite decomposition. However, a strong PbI₂ signal was found in the nPA films with the solvent volatilization rate further slowed down (Fig. 2e), which was attributed to the destruction of perovskite structure by residual solution (Fig. 2d). Despite the low polarity of the nPA, the reduction in solution evaporation rate inadvertently introduces excessive H₂O into the perovskite films, exacerbating degradation during annealing[34]. The UV-vis spectra and Tauc-plots of perovskite films fabricated using various alcohols are shown in Supplementary Figs. 9 and 10. Additionally, the UV-vis spectra of the inorganic framework are detailed in Supplementary Fig. 9. These results elucidate both the polarity and evaporation rate of the solvent

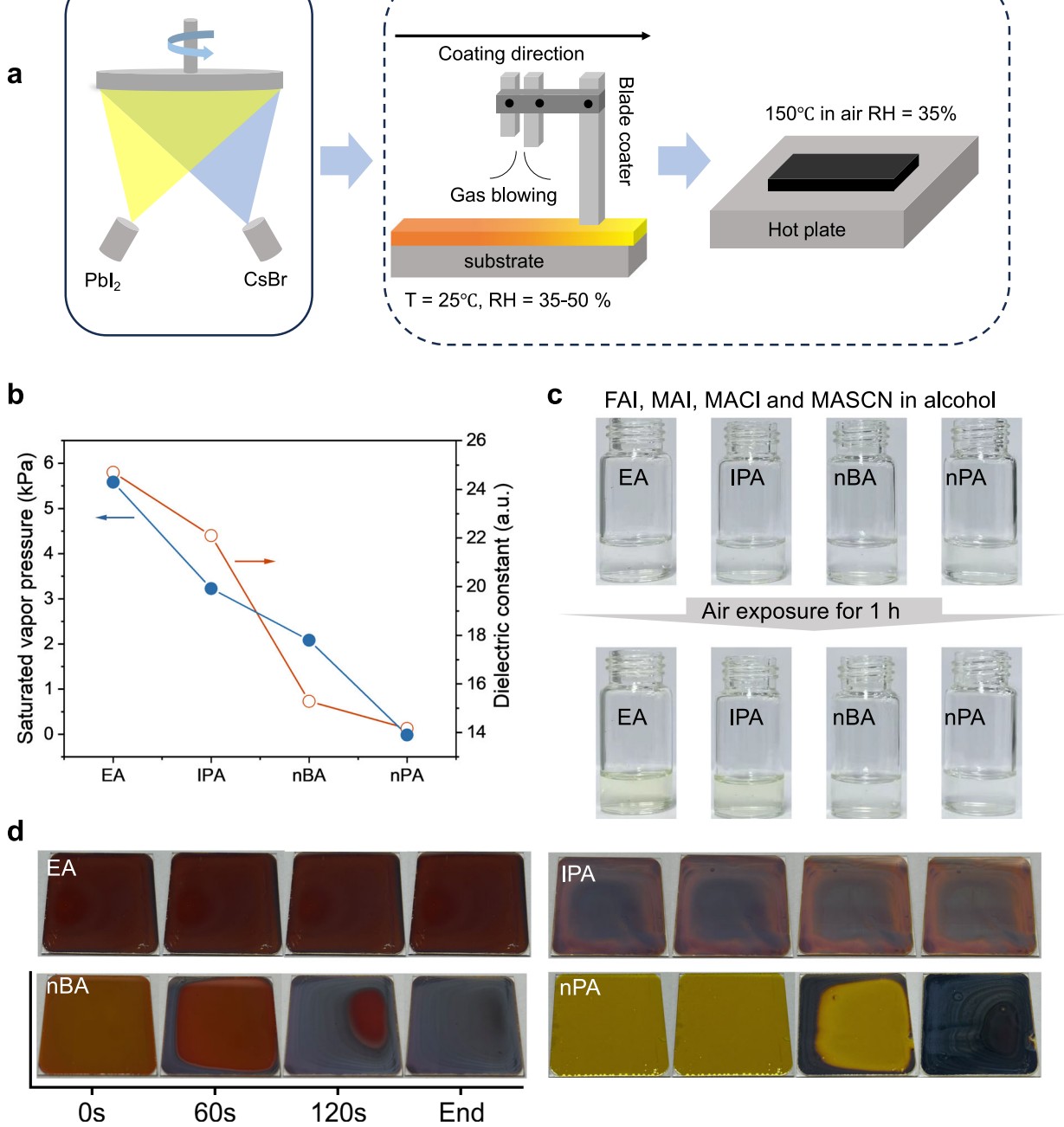

**Fig. 1 | Effect of different alcohols on perovskite films. a** Schematic of the hybrid two-step deposition method. **b** Physical parameters of different alcohols. **c** Images of organic salts used different alcohols and after exposed to air for 1 h. **d** Images of perovskite films after blade-coating organic salts without gas-quenching and annealing. The direction of blade-coating is from left to right.

have a joint effect on $H_2O$ absorption levels. In this view, nBA emerges as the optimal solvent for our specific requirements.

To discern the impact of different alcohol solvents on the defect density of perovskite layers, we performed steady-state photoluminescence (PL) measurements on samples with the configuration of glass/perovskite. As shown in Fig. 2f, for EA films, the PL emission peak of the glass side exhibited a blue shift by several nanometers relative to the others. This shift indicates a residual amount of $PbI_2$ at the bottom of the perovskite, owing to the incomplete conversion of $PbI_2$. Notably, the nBA films exhibited the highest PL intensity, surpassing both IPA and nPA films. This enhancement is attributed to the enlarged grain size and effective elimination of $PbI_2$, which in turn reduces the density of grain boundaries and suppresses the non-radiative recombination.

In addition, the time-resolved photoluminescence (TRPL) measurements further supported these findings, with the lifetime of each sample recorded at 136.3, 146.6, 350.7 and 142.9 ns, respectively (Fig. 2g). These results highlight the superior performance of nBA in minimizing non-radiative recombination within the perovskite bulk.

We then performed PL mapping test to investigate the homogeneity of the films, as shown in Fig. 2h–j. Given the significant amount of $PbI_2$ in EA films—which will notably passivate the defects and enhance the PL signal strength (as detailed in Supplementary Fig. 11)—EA films were excluded from this part of the analysis. The nBA and nPA films demonstrated superior uniformity compared to the IPA films, a trait ascribed to their lower saturated vapor pressure. This characteristic, combined with the solvent's extended chain length, leads to

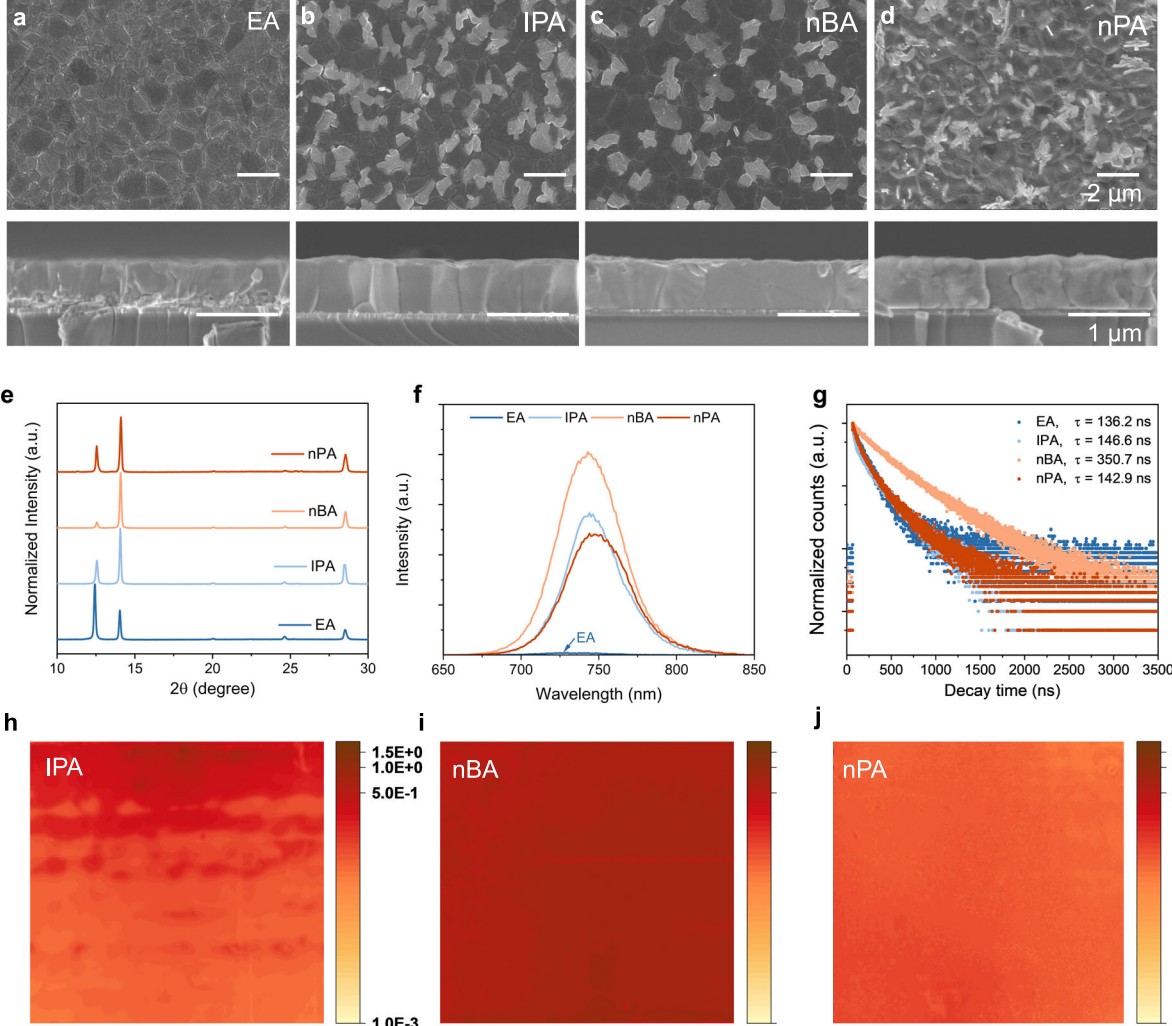

**Fig. 2 | Characterization of perovskite films fabricated by different alcohols.** a–d Top-view and cross-sectional SEM images. e XRD patterns of perovskite films after annealing. f PL spectra of perovskite films with the emission from the glass side. g Time-resolved PL transients of perovskite films. For TRPL, double exponentials were used for fitting the curves. h–j PL mapping of perovskite films with the active area for 1.5 cm ∗ 1.5 cm.

slower volatilization, while reduced polarity further restricts water ingress into the films. Both factors contribute to a diminished crystallization rate of perovskite, yielding films with enhanced homogeneity[47]. However, the slow volatilization rate of the solvent allows the residual solution to continue interacting with the perovskite through dissolution-crystallization reactions. This process tends to produce a non-optically active δ-phase and creates voids within the bulk[48], culminating in a diminished PL mapping signal in nPA films. Overall, the nBA films demonstrated less non-radiative recombination and superior uniformity, making them conducive to the scale-up fabrication of perovskite films.

**PV performance and photostability of solar cells**

We fabricated the single-junction perovskite solar cells with an architecture of Glass/ITO/NiO/SAM/1.68 eV-perovskite/$C_{60}$/SnO$_x$/Cu. The schematic structure is shown in Fig. 3a while the detailed photovoltaic parameters of the devices with an active area of 0.049 cm$^2$ applying EA, IPA, nBA and nPA are summarized in Supplementary Table 3 and Fig. 3b. For further comparison, we constructed devices under two distinct conditions: an N$_2$ environment and ambient air, with their respective photovoltaic parameters detailed in Supplementary Fig. 12. Devices fabricated in air exhibit smaller $V_{OC}$ compared to those fabricated in N$_2$ glove box, which can be attributed to moisture-induced

films deterioration. More notably, air-fabricated devices generally suffered from pronounced efficiency losses, except for those using nBA solvent. This exception highlights nBA's resilience to air exposure during fabrication, with such devices achieving the highest conversion efficiency. In our champion devices, nBA devices displayed distinct advantages in $V_{OC}$, $J_{SC}$, and FF with a narrower distribution proving its higher repeatability, as shown in Fig. 3b. According to Fig. 3c and Supplementary Table 4, the improvement of nBA devices in $V_{OC}$ and $J_{SC}$ compared with IPA groups was attributed to the lower non-radiative recombination loss and parasitic absorption caused by PbI$_2$ in the surface and bulk, which was also beneficial to the cells' light stability (as shown in Supplementary Fig. 13). The integrated $J_{SC}$ value from the external quantum efficiency (EQE) curve in Fig. 3d was calculated to be 20.81 and 20.99 mA cm$^{-2}$, respectively, corresponding well with the values obtained from J−V measurements. Compared with the IPA devices, the nBA displayed improved charge collection, particularly between 400 and 600 nm, due to the larger grain sizes minimizing recombination[49]. In order to prove the influence of uniformity on the performance of large-area devices, we compared the J−V curves of devices with a 1.044 cm$^2$ aperture area fabricated by IPA and nBA (Fig. 3e and Supplementary Figs. 16 and 17), and the specific data are shown in Supplementary Table 5. The nBA devices outperformed the IPA counterparts in terms of FF and $J_{SC}$, attributed to superior

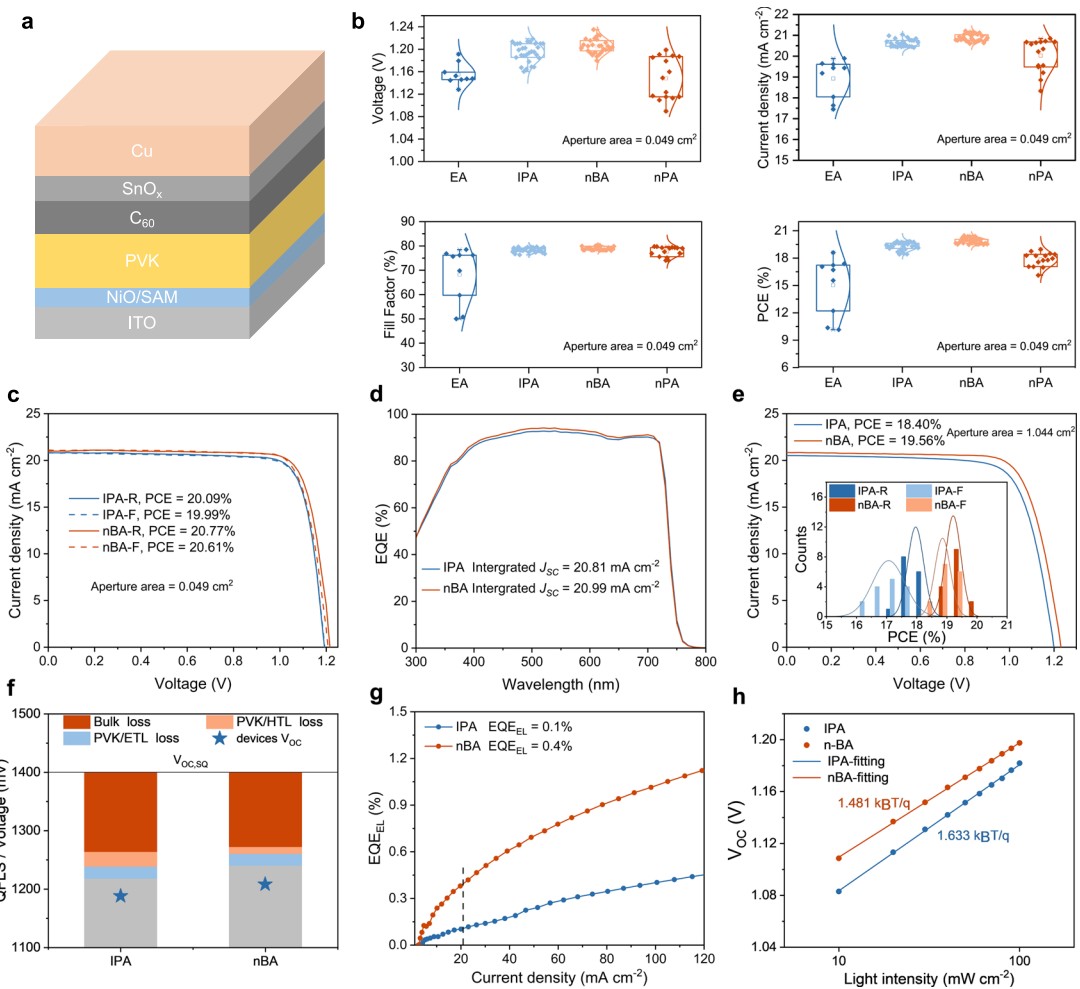

**Fig. 3 | Photovoltaic performance of PSCs fabricated by varied alcohols.**
**a** Schematic architecture of single junction. **b** Photovoltaic parameters for IPA and
nBA devices. **c** *J–V* curves of the champion opaque devices (0.049 cm² aperture
area). **d** EQE spectra of the champion device. **e** *J–V* curves of the champion opaque
devices (1.044 cm² aperture area); PCE distributions of 15 devices for each sample
are shown inset. **f** QFLS values extracted from the PL spectra for neat perovskite,
HTL/perovskite and HTL/perovskite/ETL. **g** EL spectra for IPA and nBA perovskite
devices. **h** *V*$_{OC}$ evolution as a function of light intensities for the IPA and nBA
perovskite devices.

uniformity. Additionally, from the EQE spectra of eight cells with a
small area of 0.049 cm² (Supplementary Figs. 14 and 15), we observed
that the nBA devices exhibited a much narrower distribution of the
corresponding integrated current. Furthermore, we fabricated PSCs
with an area of 1.044 cm², producing 15 devices per type. The histo-
gram of their PCE was displayed in the inset of Fig. 3e. Moreover, we
compared the photovoltaic parameters of devices fabricated by IPA
and nBA in different humidity, and the XRD of films were shown as well
(Supplementary Figs. 18 and 19), which proved that nBA hinders the
effect of moisture during the fabrication of devices.

We then carried out photoluminescence quantum yield (PLQY)
measurements to quantify the quasi-Fermi level splitting (QFLS) in the
neat perovskite layers and the stacks by different layers (Fig. 3f)[50–52].
The implied *V*$_{OC}$ values estimated from the PLQY measurements were
in good agreement with the values obtained from the *J–V* results. The
above results suggested that replacing IPA with nBA could promote
the conversion of PbI$_2$, thereby synergistically mitigating the non-
radiative recombination losses both in the bulk and in the interface
between hole-transport-layer (HTL) and perovskite. The *V*$_{OC}$, indicative
of the recombination rate within devices, was assessed through
the EQE at short circuit current conditions, effectively modeling the
device as a light-emitting diode[36]. Furthermore, under the injection
current of 21 mA cm⁻² (equal to short circuit current *J*$_{ph}$), the

electroluminescence (EL) efficiencies of IPA and nBA devices were 0.1%
and 0.4% (Fig. 3g), corresponding to the *V*$_{OC}$ loss of 0.180 and 0.144 V,
respectively. This result is almost consistent with the *J–V* results, that
is, the IPA and nBA devices showed a *V*$_{OC}$ of around 1.20 V and 1.22 V.
To further study the carrier recombination behavior, we investigated
the dependence of the *V*$_{OC}$ on the light intensity[53], as shown in Fig. 3h.
The semilogarithmic relationship displayed follows the expression
with a slope = *nkT/q* log$_{10}$e, where n is the diode quality factor. The IPA
and nBA devices exhibited *n* values of 1.633 and 1.481, respectively,
indicating reduced trap-assisted recombination in the nBA device.

Considering the need for thicker perovskite layers when
fabricating on textured silicon, relevant characterizations for per-
ovskite films on both glass and textured silicon substrates were
conducted, as shown in Supplementary Figs. 20–26. However, the
limited solvent penetration depth of IPA led to a significant amount
of unreacted PbI$_2$ in the underlying layer and further degraded the
performance of the devices. While complete conversion of the
inorganic framework to perovskite is achievable through adjust-
ments in parameters like quenching gas pressure and blade-coating
rate[54,55], such modifications can detract from film uniformity and
device performance, as evidenced in Supplementary Figs. 27–30.
Consequently, parameter tuning was not utilized to fully convert IPA
films in tandem devices.

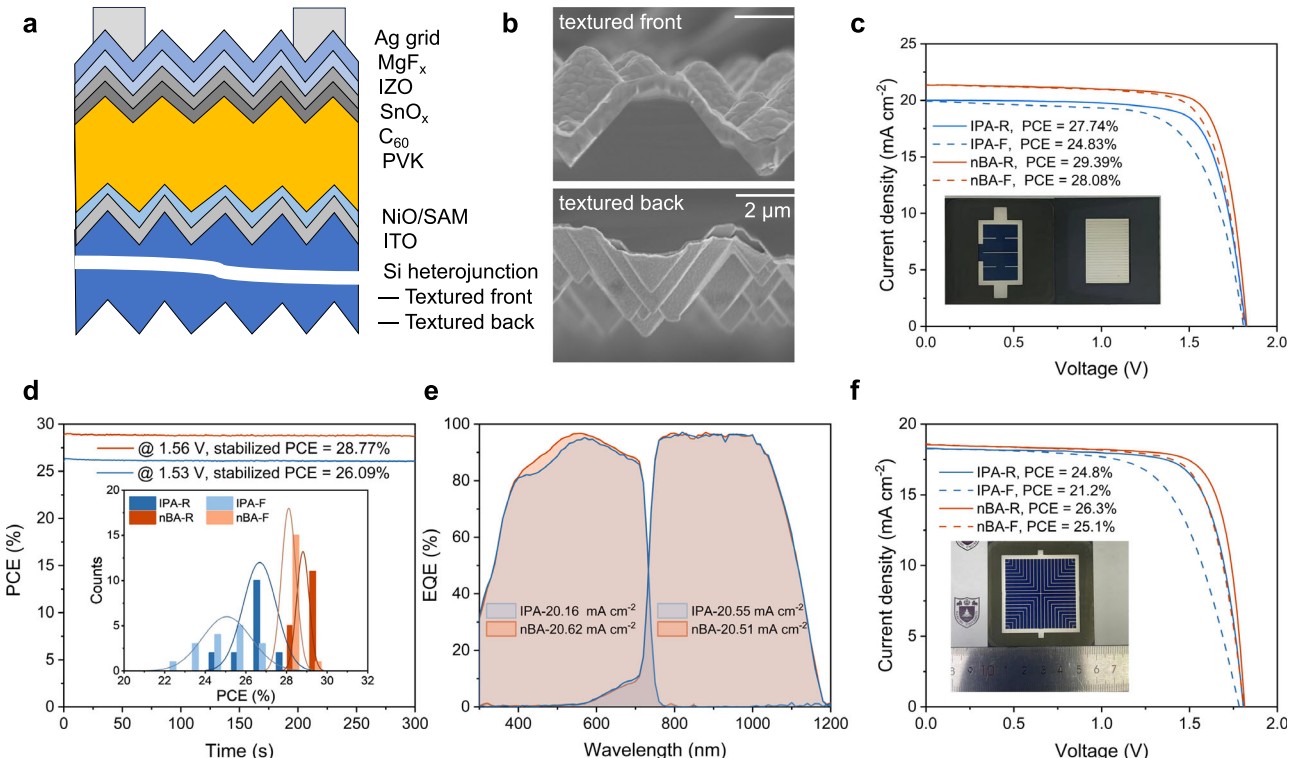

**Fig. 4 | The device characterization of fully textured perovskite/SHJ tandems.**
**a** Schematic diagram of perovskite/SHJ tandem solar cell. **b** Cross-sectional SEM images of perovskite/SHJ (average pyramid size is 2–3 μm) tandem for nBA devices. **c** *J*–*V* curves of the tandem device (1.044 cm² aperture area); the digital photo of a device is shown in the inset. **d** MPP tracking of the tandems; PCE distributions of 16 individual tandem devices for each type is shown in the inset. **e** EQE spectra of a current-matched fully textured monolithic perovskite/SHJ tandem cell. **f** *J*–*V* curves of the tandem device (16 cm² aperture area); the digital photo of a device is shown in the inset.

Specifically, the illustration of the tandem device is demonstrated in Fig. 4a and a broader area showing the top-view as well as cross-sectional SEM images of the bottom SHJ is seen in Supplementary Fig. 31. The performance of SHJ cell with and without semitransparent perovskite as a filter is shown in Supplementary Fig. 32 and Supplementary Table 6. It can be clearly seen from Fig. 4b that the textured surface with pyramid sizes of 2–3 μm was well-covered by the conformally coated perovskite films as well as other functional layers. The corresponding device performance is depicted in Fig. 4c and d; a tandem solar cell with an active area of 1.044 cm² achieved a champion PCE of 29.4% ($V_{OC}$ = 1.83 V, $J_{SC}$ = 20.45 mA cm⁻² and FF = 78.63%) under reverse scan and the stabilized PCE was observed to be 28.8%. Moreover, an independently certified efficiency of 28.7% was tested from Fraunhofer ISE (shown in Supplementary Fig. 33).

As shown in Fig. 4d, the integrated $J_{SC}$ value of the front and back subcell from EQE spectra (Fig. 4e) was 20.62 and 20.51 mA cm⁻², respectively, which was in good agreement with the $J_{SC}$ value determined from the *J*–*V* measurements considering the loss caused by Ag grid. We further evaluated the operation stability of encapsulated tandem solar cells by measuring the maximum power output under 1-sun-equivalent illumination in ambient air with a relative humidity of 30–50%. The encapsulated device retained 96.8% of its initial PCE after 780 h of maximum power point (MPP) tracking (Supplementary Fig. 34).

To validate the applicability of our approach for scalable fabrication, we applied blade-coating to produce perovskite films on a 36 cm² glass substrate. Subsequent steady PL and XRD tests conducted on samples from different regions of perovskite films (Supplementary Figs. 35 and 36) demonstrated superior uniformity in nBA films compared to IPA films. Furthermore, we fabricated 36 cm² perovskite/silicon tandem cells (aperture area, 16 cm²) and achieved a conversion efficiency of 26.3% ($V_{OC}$ = 1.815 V, $J_{SC}$ = 18.54 mA cm⁻², FF = 78.31%), which is among the highest PCE of large-area perovskite/silicon tandem cells[11]. The consistency of the EQE spectra at different regions suggested that the film exhibited excellent uniformity (Supplementary Fig. 37).

For the further development of perovskite/silicon tandem solar cells, scaling up the size of the perovskite films to M6 (166 mm*166 mm) becomes essential, a goal that proves challenging with blade-coating due to issues with film uniformity. Therefore, slot-die coating, an expandable technology that allows continuous liquid injection, emerges as the preferable future method[49,56]. Critical to this method is the complete conversion of the inorganic framework into perovskite films, achievable through careful adjustment of precursor solution concentration, the injection rate of the precursor solution, the rate of slot-die, gap distances between the blade and substrate, and quenching gas pressure. Digital photographs of the perovskite films fabricated under these conditions are depicted in Supplementary Figs. 38–40. With the optimum slot-die coating parameters (1 mL/min, 100 μm and 30 PSI), we achieved perovskite films with excellent homogeneity (Supplementary Fig. 41). The optimal device delivered a PCE of 25.9% for 16 cm² ($V_{OC}$ = 1.823 V, $J_{SC}$ = 18.50 mA cm⁻², FF = 76.63%), as shown in Supplementary Fig. 42. These results are anticipated to surpass the efficiency of devices fabricated by the blade-coating in the future.

## Discussion

In summary, we found that the moisture in air is involved in the reaction of the organic salt and the inorganic frame, which adversely affected the perovskite films and devices. Therefore, we innovatively proposed that the polarity and volatilization rate of the solvent together affect the water absorption, and selected nBA as the optimal organic salt solvent, which can also promote the uniformity of the

perovskite films. As a result, a single-junction perovskite solar cell with bandgap of 1.68 eV achieved a champion PCE of 20.79% (0.049 cm$^2$ aperture area) and 19.56% (1.044 cm$^2$ aperture area). This tandem yields a certified efficiency of 28.7% for 1.044 cm$^2$ and a laboratory efficiency of 26.3% for 16 cm$^2$. The encapsulated devices retained 96.8% of their initial output after 780 h of MPP tracking. Furthermore, our demonstration of tandem solar cells fabricated using slot-die coating achieved a conversion efficiency of 25.9% for 16 cm$^2$. This advance provides a new solution for the large-scale practical production of perovskite/silicon tandem solar cells, which is beneficial to realize the commercialization of perovskite/silicon tandem solar cells.

## Methods

### Materials

All materials were used as received without further purification. The organic salts (FAI, FABr, MASCN and CF3-PACl with the purity of > 99%) were purchased from GreatCell Solar Materials. MACl was purchased from Xi'an Polymer Light Technology. PbI$_2$ (99.99%, trace metals basis) and CsBr (99.9%), 2PACz (>98.0%) and MeO-2PACz (>98.0%) were purchased from Tokyo Chemical Industry (TCI). EA (99.8% anhydrous), IPA (99.5% anhydrous), nBA (99.5% anhydrous) and nPA (99.0%) were purchased from Sigma-Aldrich. The C$_{60}$ was purchased from Nano-C. MgF$_2$ (99.999%) was purchased from ZhongNuo Advanced Material Technology Co., Ltd. Tetrakis (dimethylamino) tin(iv) (99.9999%) was purchased from Nanjing Ai Mou Yuan Scientific Equipment.

### Organic salt solution

In the fabrication of single-junction solar cells, FAI (2.8 mmol), FABr (2.8 mmol), MACl (0.7 mmol) and MASCN (0.7 mmol) were dissolved in 15 ml alcohol, such as EA, IPA, nBA and nPA. In the fabrication of tandem solar cells, FAI (3.36 mmol), FABr (3.36 mmol), MACl (0.84 mmol) and MASCN (0.84 mmol) were dissolved in 15 ml alcohol, such as IPA and nBA.

### Fabrication of single-junction perovskite solar cells

Patterned ITO glass substrates were sequentially cleaned using ultrasonication in acetone and isopropanol for 30 min, respectively. The cleaned ITO substrates were treated with UV ozone for 15 min, followed by sputtering a NiO hole transport layer on the substrates using a 3.5-inch NiO target (Plasmaterials, 99.9%) at 150 °C in an Ar atmosphere. The NiO layer was sputtered at a rate of 0.2 Å s$^1$ using a 90 W of RF power. Subsequently, the solutions of 2PACz and MeO-2PACz with the same concentration (1 mmol L$^{-1}$ in IPA) were mixed with volume ratios (75:25) and then were spin-coated on the NiO film at 4000 r.p.m. for 20 s, followed by annealing at 100 °C for 5 min in air to form molecule bridged interfaces, followed by washing with IPA for 2 times. The perovskite was produced using a two-step process. First, PbI$_2$ and CsBr were co-evaporated by thermal evaporation. The evaporation rate of PbI$_2$ and CsBr was 4 and 0.8 Å s$^{-1}$, respectively. The thickness of the inorganic frame is 405 nm on a glass substrate and 540 nm on a textured silicon substrate. Following this, for small-area devices (2.5 cm * 2.5 cm) 25 μL organic salt solution was blade-coated with a rate of 20 mm s$^{-1}$ assisted with N$_2$ in air; the gap between blade and substrate is 250 μm and the pressure of the N$_2$-knife is 30 PSI. As for large-area devices (6 cm * 6 cm), the amount of solution was increased to 60 μL. The ambient temperature is 25 °C and the humidity is 35–50%. Next, the films were transferred to a hotplate and annealed at 150 °C in air (35% RH) for 20 min. Next, the substrates were transferred to the evaporation system. For the device with passivation, the substrates were transferred into glovebox to spin-coating 1 mg ml$^{-1}$ CF3-PACl with 4000 r.p.m. for 30 s and annealed at 100 °C for 10 min. Sequentially, 20 nm of C$_{60}$ film was deposited by thermal evaporation at a rate of 0.2 Å s$^{-1}$. The substrates were then transferred to the ALD system (Veeco Savannah S200) to deposit 15 nm SnO$_x$ at low temperature (typically 100 °C) using precursors of tetrakis

(dimethylamino) tin(iv) and deionized water. After that, 150 nm of Cu as an electrode was deposited by thermal evaporation at the rates of 1.0 Å s$^{-1}$. For the semitransparent perovskite solar cells, 80 nm of IZO was sputtered onto the stack instead of the deposition of Cu electrodes. Ag fingers (100 nm) were deposited by thermal evaporation at the rate of 1.0 Å s$^{-1}$. Finally, 100 nm of MgF$_2$ as an antireflection layer was deposited by electron beam evaporation.

### Silicon bottom solar cell fabrication

Silicon heterojunction (SHJ) bottom cells were fabricated on n-type double-side textured silicon wafers (CZ, 4 inch, 1–5 Ω resistivity, 260 μm thickness). The intrinsic, n-type and p-type hydrogenated amorphous silicon layers were deposited on both sides of the wafer using plasma-enhanced chemical vapor deposition (PECVD). The back contact of the silicon cells was fabricated by stacking sputtered ITO (80 nm) and then thermal evaporated Ag (150 nm).

### Perovskite/silicon monolithic tandem fabrication

Before the fabrication of perovskite top cells, the 4-inch silicon wafers were laser cut to 2.5 * 2.5 cm$^2$ substrates. Then the silicon substrates were annealed at 200 °C for 10 min to recover the sputtering damage. Next, 20 nm ITO was sputtered as a recombination junction. The perovskite top cells were fabricated using the identical process of the semitransparent perovskite solar cells.

### Characterization of solar cells

For tandem solar cells, the current density-voltage (J–V) characteristics were measured using a Keithley 2400 source meter under the illumination of the solar simulator (EnliTech, Class AAA) at the light intensity of 100 mW cm$^{-2}$ as checked with NREL calibrated reference solar cells (KG-5 and KG-0 reference cells were used for the measurements of wide-bandgap and narrow-bandgap solar cells, respectively). Unless otherwise stated, the J–V curves were all measured in air with a scanning rate of 100 mV s$^{-1}$ (voltage steps of 20 mV and a delay time of 100 ms). The active area was determined by the aperture shade masks (0.049 cm$^2$ and 1.044 cm$^2$) placed on the glass side of the solar cells. EQE measurements were performed in ambient air using a QE system (EnliTech) with monochromatic light focused on device pixels and a chopper frequency of 20 Hz. For tandem solar cells, the J-V characteristics were carried out under the illumination of a two-lamp high spectral match solar simulator (SAN-EI ELECTRIC, XHS-50S1). EQE measurements were performed in ambient air, and the bias illumination from highly bright LEDs with emission peaks of 850 and 460 nm was used for the measurements of the front and back subcells, respectively. No bias voltage was applied during the EQE measurements of tandems.

### Stability tests of solar cells

The operational stability tests were carried out under full AM1.5 G illumination (multi-colored LED solar simulator, 100 mW cm$^{-2}$) using a home-build LabVIEW-based MPP tracking system in ambient conditions with humidity of ~40%. The modules were encapsulated with a cover glass and butyl rubber, POE padding is used in the middle. They were melted at 100 °C and applied pressure to expel air. The dark long-term shelf stability assessments of modules were carried out by repeating the J-V characterizations over various times, and the devices were stored in ambient conditions with humidity of 40 ± 5%.

### Other characterizations

SEM images were obtained using a TESCAN microscope with an accelerating voltage of 2 kV. XRD patterns were collected using a Bruker D8 Advance equipped with a NaI scintillation counter and using monochromatized Copper Kα radiation ($\lambda = 1.5406$ Å). Photoluminescence (PL) was measured using a HORIBA Nanolog fluorescence spectrometer (Horiba Instruments) with an excitation

wavelength of 485 nm. The intensity of the laser was adjusted to 1-Sun equivalent. TRPL was measured using a Horiba Fluorolog-3 time-correlated single-photon counting system; the samples were excited using a pulsed laser with a wavelength of 485 nm from the perovskite side. The PL decay curves were fitted with a single exponential function for perovskite films fabricated on glass. The PL decay curves were fitted with biexponential function to obtain the fast and slow PL decay lifetimes of $\tau_1$ and $\tau_2$ and the corresponding coefficients of $A_1$ and $A_2$ of perovskite films, respectively. Then the PL effective decay lifetime $\tau_{eff}$ was calculated by the following equation: $\tau_{eff} = (A_1\tau_1 + A_2\tau_2) / (A_1 + A_2)$. The PLQY was obtained via exciting the corresponding samples with a 405 nm laser inside an integrated sphere (Newport, 70682NS). The signal was coupled into a fiber and collected by an Ocean Optics QEPro spectrometer. The integrating sphere and fiber were calibrated using a lamp of the known spectrum (Ocean Insight, HL-3P-INT-CAL). The illumination intensity was modulated with neutral density (ND) filters to achieve a near 1-sun equivalent photon flux at a bandgap of 1.8 eV. The PL imaging tool is an LIS-R1 system sourced from BT Imaging. The samples were excited with a continuous-wave 532 nm laser (one-sun on-sample intensity). The camera was a $1024 * 1024$ pixels, charge-coupled-device silicon detector with a detection range of 400–1060 nm.

## Reporting summary

Further information on research design is available in the Nature Portfolio Reporting Summary linked to this article.

## Data availability

Source data are provided with this paper. The main data supporting the findings of this study are available within the published article and its Supplementary Information and source data files. Additional data are available from the corresponding author on request. Source data are provided with this paper.

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

## Acknowledgements

This work was financially supported by the National Key R&D Program of China (2023YFB4202501), National Natural Science Foundation of China (U21A2076 and 61974063), Natural Science Foundation of Jiangsu Province (BK20230790, BE2022021, BE2022026, BK20202008, BK20190315), Fundamental Research Funds for the Central Universities (0213/14380206; 0205/14380252), Frontiers Science Center for Critical Earth Material Cycling Fund (DLTD2109), and Program for Innovative Talents and Entrepreneur in Jiangsu. The research has also received funding from China Postdoctoral Science Foundation (2022M712341), Jiangsu Planned Projects for Postdoctoral Research Funds (2021K084A), and the Science and Technology Support Plan (Industrial) Project of Changzhou City (CE20220032). The authors thank Fraunhofer ISE for the third-party certification report.

## Author contributions

H.T. conceived and directed the overall project. X.Z., W.K. and J.W. fabricated all the devices and conducted the characterization. J.H., H.L., Z.H., X.L., Z.L., H.S., Y.W., C.L., P.W., H.G., M.L., R.L. and L.L. helped with the device fabrication and material characterization. A.D.B. and H.T.N performed the PL imaging characterization. R.X., H.Li., Y.M., X.Z., G.Y., Y.C., Z.F. and J.G. helped with the fabrication of SHJ. X.Z., W.K., J.W., J.G. and H.T. wrote the manuscript. All authors read and commented on the manuscript.

## Competing interests

Hairen Tan is the founder, Chief Scientific Officer, and Chairman of Renshine Solar Co., Ltd., a company that is commercializing perovskite PVs. The remaining authors declare no competing interests.
