## [Peer Review File · Nature Communications]

Solvent Engineering for Scalable Fabrication of Perovskite/Silicon Tandem Solar Cells in AirREVIEWER COMMENTS

Reviewer #1 (Remarks to the Author):

The work of Zheng et al. addresses an important topic for the large-scale processing of perovskite/silicon tandem solar cells, which is rarely reported but is quite critical for the development of the field. From this perspective, I find the topic relevant to the wider community. The influence of the solvent on hybrid perovskites is not a new topic, and several previous works have reported on adjusting the boiling point of the solvent to control the evaporation rate, which directly influences crystallization. However, solving this issue with a single solvent sounds easy to adopt for industry. From this perspective, the article offers new insights. The characterizations provided in the article are mostly supportive. However, the rationale behind the choice of the technique and its transition from spin coating could be explained better, as elaborated below. The device performances are impressive, considering those devices are processed in ambient air and with a solvent sweeping method. Overall, I believe that the article will bring new knowledge to the field. However, I also believe that the article can benefit from the recommendations below to further improve its quality.

Major Comments:

1. Indeed, blade coating is a promising starting point for scaled processing, specifically in ambient air. However, I wonder what challenges the authors envision for the transformation to slot-die coating. Incorporating the authors' insights into the article would be interesting for follow-up works. For instance, what challenges did they face, how precise is the parameter control for the repeatability of the processes, etc.? Additionally, insights into why the authors considered the evaporation/blade coating technique over others (e.g., gas reaction conversion) could be of interest. More elaboration would be beneficial. The authors are recommended to cite the proper literature when explaining their choice.
2. Can the authors provide more details about the perovskite structures in Figure 1a? For instance, can they show the scaffold of PbI_2/CsBr ? What was the thickness of this scaffold? Also, what are those whitish colors in Figure 2b-d? Are they remnants of the scaffold or PbI_2 ? I extend this question to the textured solar cells; can the authors show the scaffold for that structure as well?
3. Related to the previous question, although the authors reported in their previous works that non-nBA-based solvents were giving fully converted perovskites, what changed here, and why did the structure become non-converted? This question basically targets whether this is due to the change of the scaffold or not.

4. As the authors focus on the evaporation rate of the solvent, I wonder if it would be possible to tune the blading speed to achieve similar results.

5. The authors are encouraged to provide a full report of the certification.

6. According to Figure S1, processing the perovskites in a nitrogen environment is the best condition for IPA. For the rest of the work, the authors investigate different solvents for ambient air. I think throughout the article, N₂ conditions would be a good reference. If this makes a huge difference, there might be a cost-effective solution to integrate the processing in a protected environment. If there is no significant difference, this is already important information for the literature.

7. There is an MASCN additive inside the precursor solution, which does not seem like a standard salt. How critical is this? How id the entire process influenced from this (solvent evaporation rate, crystallization control, etc.)?

Minor Comments:

1. In the introduction, the authors claim, "Nevertheless, perovskite films with a 1 μm thickness employing...". I wonder why the thickness should be 1 μm?

2. In Figure 1c, can the authors make clear which organic salts were included in the solution?

3. What is the scan direction in the images in Figure 1d?

4. Figure 1 requires more details such as the deposition technique.

5. Can the authors confirm that all samples in Figure 2 were made in the atmosphere (at what relative humidity) and blade coating?

6. Can the authors add the absorptance of the PbI₂ to Figure S5, as they claim PbI₂ causes parasitic absorption?

7. Can the authors show the data points in Figure 3g?
8. In Figure 4d, which device's statistical results are reported?
9. Can the authors provide the substrate temperature during blading and the amount of solvent/substrate area, which will increase the precision of their recipe?
10. Is there any specific reason that the authors use a high evaporation rate of 4 and 0.8 Å s⁻¹ for PbI₂ and CsBr?
11. A batch of devices showing the comparison of blade/spin coating processes in ambient and N₂ conditions with different solvents would support the claims of the article, specifically the influence of moisture.
12. Can the author explain why the overall tandem FF is higher than the bottom cells (Table S6), although the cells are current-matched?
13. Can the authors give reverse and forward scans together in Figure 4c and f? The same for Figure 3e.
14. The authors have a strong emphasis on humidity ingress. I wonder what the water content of the solvents is they use. Are they anhydrous? Is this important?
15. It seems 26.3% is not the highest PCE reported for large-area perovskite silicon tandems according to the results in this article (DOI: 10.1126/science.adh3849).
16. Having page and line numbers would be useful to provide feedback.
17. There are several typos in the article, and the authors are recommended to check and correct them.

Reviewer #2 (Remarks to the Author):

In the paper “Solvent Engineering for Scalable Fabrication of Perovskite/Silicon Tandem Solar Cells in Air”, the authors report a new solution for wide-bandgap perovskite solar cells and the large-scale practical production of perovskite/silicon tandem solar cells based on industrial pyramid (2-3 μm) silicon heterojunction fabricated in air. They innovatively proposed that the polarity and volatilization rate of

solvent together affect the water absorption. This manuscript represents a significant advance in the field of perovskite/silicon tandem solar cells. The conclusions are reasonable and supported by the experimental evidences. I recommend publication after revision:

1. In the paper, authors presented “highly polar alcohols are susceptible to environmental moisture”. The conclusion is unwarranted. Please explain the reasons in details or add some references.
2. In Fig. 1c, the color of EA and IPA solutions turned from colorless to light yellow after 1 hour of air exposure. What is the “yellow”? The author should explain the reason clearly to understand the relation between moisture and polarity of alcohol solutions.
3. The author should present more explanations to support the description of “the initial perovskite structure was destructed by residual solution in nPA films, resulting in the formation of white circular stripes”. And, the author should make it clear that what is “white circular stripes”?
4. The author mentioned that “the EA film contained a large amount of PbI_2 , which would remarkably passivate the defects and increased strength of PL signal”. Why does the intensity of PL of EA films in Fig. 2f not increase?
5. The PCE of fabricated 36 cm^2 perovskite/silicon tandem cells (aperture area, 16 cm^2), of 26.3% is the highest PCE of large-area perovskite/silicon tandem cells reported. The author should add a table to summarize and make a comparison with other large-area perovskite/silicon tandem cells reported.
6. In Fig. 4e, the EQE of nBA is higher than IPA between 400-600 nm. The author should explain the reason.
7. Why thicker perovskite layers are needed when fabricating on textured silicon even by two-step sequential deposition method? I noticed that the thickness of perovskite is $\sim 1 \mu\text{m}$. How do you determine the thickness of perovskite in tandem solar cells?
8. The author adjusted the pressure of the N_2 knife. And, the according literature, Joule 6, 1–13, August 17, 2022, should be cited.
9. The spelling mistakes should be corrected, such as, “siliocn”.

Reviewer #3 (Remarks to the Author):

In the manuscript titled with “Solvent Engineering for Scalable Fabrication of Perovskite/Silicon Tandem Solar Cells in Air”, the authors reported the efficiency of 29.4% for perovskite/silicon tandem solar cells and 26.4% for an aperture area of 16 cm². The authors think nBA solvent can mitigate the impact of moisture in air due to the low polarity and moderated volatilization rate. However, how the different polarities of alcohols impact the conversation speed of PbI₂ and the uniformity of perovskite film, the detailed interaction mechanism is unclear. I would recommend to accept it once the authors can address the following comments below.

1. Compared with IPA solvent, nBA solvent can mitigate the impact of moisture in air. The efficiency of nBA device is higher than that of IPA device when the solar cell devices were fabricated in air. However, the authors should give the photovoltaic performances of nBA and IPA devices when the solar cell devices were fabricated in N₂ environment. Whether nBA solvent can be used in N₂ environment with better performance?
2. As shown in Figure 1c, why the color of EA and IPA solutions turned to light yellow, the authors should give more detailed explanation about what factors contribute to the change in color and how the reaction occur?
3. The conversation and crystallinity of perovskite films via two-step method mainly depends on the solvent volatilization rate. How the different polarities of alcohols impact the conversation speed of PbI₂ and the uniformity of perovskite film, the authors should give more detailed explanation.
4. As shown in Figure 2d, nPA films have been destroyed by residual solution. However, Figure 2f shows the highest PL intensity of nPA films. The enlarged grain size of nPA films can't be clearly observed in SEM image. Moreover, there are large amount of PbI₂ existed on the nPA film surface. The authors should give more related evidence to prove the increased PL intensity of nPA films.

Point-to-point response to the Reviewers' comments

Manuscript #: NCOMMS-24-01196

We sincerely thank all reviewers for their much-valued suggestions, which have enabled us to substantially improve the manuscript's quality. Followings are the detailed actions taken in light of reviewers' comments.

Reviewer #1 (Remarks to the Author):

The work of Zheng et al. addresses an important topic for the large-scale processing of perovskite/silicon tandem solar cells, which is rarely reported but is quite critical for the development of the field. From this perspective, I find the topic relevant to the wider community. The influence of the solvent on hybrid perovskites is not a new topic, and several previous works have reported on adjusting the boiling point of the solvent to control the evaporation rate, which directly influences crystallization. However, solving this issue with a single solvent sounds easy to adopt for industry. From this perspective, the article offers new insights. The characterizations provided in the article are mostly supportive. However, the rationale behind the choice of the technique and its transition from spin coating could be explained better, as elaborated below. The device performances are impressive, considering those devices are processed in ambient air and with a solvent sweeping method. Overall, I believe that the article will bring new knowledge to the field. However, I also believe that the article can benefit from the recommendations below to further improve its quality.

Major Comments:

1. Indeed, blade coating is a promising starting point for scaled processing, specifically in ambient air. However, I wonder what challenges the authors envision for the transformation to slot-die coating. Incorporating the authors' insights into the article

would be interesting for follow-up works. For instance, what challenges did they face, how precise is the parameter control for the repeatability of the processes, etc.? Additionally, insights into why the authors considered the evaporation/blade coating technique over others (e.g., gas reaction conversion) could be of interest. More elaboration would be beneficial. The authors are recommended to cite the proper literature when explaining their choice.

Response: We thank the Reviewer for your recognition of our work and for stimulating us to improve the manuscript.

In light of reviewer's suggestion, we have investigated the adoption of slot-die coating as an alternative to blade-coating, conducting a comparative analysis under various parameters. Via learning from experience of blade coating, we extended the same solution concentration and blade-coating rate, and explored the injection rate of precursor solution, the gap between blade and substrate, and gas pressure in order to obtain uniform perovskite films. Through this approach, we demonstrated a perovskite/silicon tandem cell with a PCE of 25.9% for 16-cm² devices. We now add these finding and improve the discussion in the revised manuscript as follows:

*“As the further development of perovskite/silicon tandem solar cells, scaling up the size of the perovskite films to M6 (166 mm*166 mm) becomes essential, a goal that proves challenging with blade-coating due to issues with film uniformity. Therefore, slot-die coating, an expandable technology that allows continuous liquid injection, emerges as the preferable future method^{50,57}. Critical to this method is the complete conversion of the inorganic framework into perovskite films, achievable through careful adjustment of precursor solution concentration, injection rate of the precursor solution, the rate of slot-die, gap distances between the blade and substrate, and quenching gas pressure. Digital photographs of the perovskite films fabricated under these conditions are depicted in **Supplementary Fig. 38-40**. With the optimum slot-die coating parameters (1 mL/min, 100 μm and 30 PSI), we achieved perovskite films with excellent homogeneity (**Supplementary Fig. 41**). The optimal device delivered a PCE*

of 25.9% for 16 cm² ($V_{oc} = 1.823$ V, $J_{sc} = 18.50$ mA cm⁻², $FF = 76.63\%$), as shown in **Supplementary Fig. 42**. These results are anticipated to surpass the efficiency of devices fabricated by the blade-coating in the future.”

Supplementary Fig. 38 Images of perovskite films fabricated using different quenching gas pressure.

Supplementary Fig. 39 Images of perovskite films fabricated with different injection rate of ink.

Supplementary Fig. 40 Images of perovskite films fabricated under different gap distances between the blade and substrate.

*Supplementary Fig. 41 XRD patterns of perovskite films (6 cm * 6 cm) from different regions.*

Supplementary Fig. 42 J-V curve of the perovskite/silicon tandem solar cell (16 cm²)

Then, we have compared the various fabrication strategies for perovskite films including the solution deposition method, the evaporation deposition method, and the hybrid evaporation and solution method. We now improve the discussion in the revised manuscript.

“However, the conventional one-step solution method for depositing perovskite films struggles to achieve full coverage over such pyramid tips due to the limited thickness (less than 1 μm), which is essential for effective carrier transport, thus raising the risk of current shunts^{7,8}.”

“To address this issue, methods such as full evaporation method (co-evaporation or sequential evaporation) and a hybrid two-step deposition method (combining evaporation of inorganic framework and solution processing of organic salts) have been developed. These methods enable the conformal deposition of perovskite films on industrially textured silicon solar cells^{8–10}. Despite these advancements, the full evaporation method faces difficulties in incorporating additive engineering strategies, vital for enhancing the performance of perovskite solar cells, which results in relatively lower performance compared with the hybrid two-step deposition¹¹. Up to now, the record efficiency of perovskite/silicon tandem solar cells fabricated via hybrid method stands at 31.25%¹². However, this impressive result has limited on lab-scale ($\sim 1 \text{ cm}^2$) devices, fabricated under N_2 environment using spin-coating,^{8,12–18} which is not conducive to scalable production^{19,20}. ”

2. Can the authors provide more details about the perovskite structures in Figure 1a? For instance, can they show the scaffold of PbI_2/CsBr ? What was the thickness of this scaffold? Also, what are those whitish colors in Figure 2b-d? Are they remnants of the scaffold or PbI_2 ? I extend this question to the textured solar cells; can the authors show the scaffold for that structure as well?

Response: In light of reviewer’s suggestion, we now include the scanning electron microscopy (SEM) images of the inorganic frameworks deposited on glass and silicon

substrates in the revised manuscript:

Additionally, the corresponding thickness of the PbI_2/CsBr scaffold have labeled on these images.

“As depicted in Fig. 1a, our process combines co-evaporation and blade-coating techniques to meet the requirements for large-area fabrication of the perovskite films. Supplementary Fig. 2 shows the deposition of inorganic framework on both glass and textured silicon substrates.”

Supplementary Fig. 2 a, b, Top-view and cross-sectional SEM images of inorganic framework fabricated on glass a, b, and on textured silicon c, d.

The appearance of a whitish color substance is attributed to the formation of PbI_2 during the air annealing, caused by the decomposition of the perovskite films (DOI: 10.1126/science.abp8873). To substantiate this assertion, the XRD patterns of the IPA perovskite films, both pre- and post-annealing, were examined and presented in Figure R1.

Fig. R1 X-ray diffraction (XRD) patterns of perovskite films before and after annealing

3. Related to the previous question, although the authors reported in their previous works that non nBA-based solvents were giving fully converted perovskites, what changed here, and why did the structure become non-converted? This question basically targets whether this is due to the change of the scaffold or not.

Response: We reported that the introduction of PbCl_2 could promote the conversion of inorganic frameworks to perovskite films in N_2 environment (doi.org/10.1021/acseenergylett.3c02002). However, the challenge in achieving fully converted perovskites in this study is different from our previous findings with non-nBA-based solvents. As shown in Figure R2, the introduction of PbCl_2 caused a degradation of performance in air-processed blade-coated devices. Consequently, we determined that PbCl_2 is not a viable additive and seek for other solutions.

Fig. R2 Photovoltaic parameters of devices fabricated with and w/o the introduction of Cl

4. As the authors focus on the evaporation rate of the solvent, I wonder if it would be possible to tune the blading speed to achieve similar results.

Response: We appreciate the reviewer's comment. In our initial experiments, we explored the effect of the blade-coating rate and gas quenching pressure on film morphology in the isopropyl alcohol (IPA) scheme, with SEM images provided in Supplementary Fig. 29. We observed that increasing the blade-coating rate facilitated the complete conversion of the inorganic framework to perovskite. However, at higher coating rates, the films began to detach from the underlying silicon substrate, adversely affecting tandem cell performance.

Further experimentation in WBG PSCs showed that device performance improved with a blade-coating rate between 10-20 mm/s, due to the full conversion of perovskite. However, within the range of 20-30 mm/s, the device performance declined due to poor contact between the perovskite film and the hole transport layer, leading to

increased voltage and FF losses. We now include the detailed discussion in the revised manuscript.

*“While complete conversion of the inorganic framework to perovskite is achievable through adjustments in parameters like quenching gas pressure and blade-coating rate^{55,56}, such modifications can detract from film uniformity and device performance, as evidenced in **Supplementary Figs. 27-30**. Consequently, parameter tuning was not utilized to fully convert IPA films in tandem devices.”*

Supplementary Fig. 27 Images of IPA films fabricated using different quenching gas pressure.

Supplementary Fig. 28 Photovoltaic parameters of IPA devices fabricated using different quenching gas pressures.

Supplementary Fig. 29 Cross-sectional SEM images of perovskite films fabricated on textured silicon with different blade-coating rate.

Supplementary Fig. 30 Photovoltaic parameters of IPA devices fabricated using different blade-coating rate

5. The authors are encouraged to provide a full report of the certification.

Response: We thank the reviewer for the suggestion. We made revision as the reviewer suggested and added the full report of the certification in the update manuscript.

Fraunhofer ISE CalLab PV Cells

Heidenhofstr.2

79110 Freiburg

Werkskalibrierschein
Proprietary calibration report

10002070TSE0523

Gegenstand Object	monofacial multi-junction solar cell
Hersteller Manufacturer	
Typ Type	PSC/Si
Fabrikat/Serien-Nr. Serial number	TSE002 / 2
Auftraggeber Customer	NanJing University/ Trina solar Co., Ltd. No.2 Tianhe Road,Trina PV Industrial Park ,Xinbei district 213032 Changzhou, Jiangsu province China
Auftragsnummer Order No.	070TSE0523
Anzahl der Seiten Number of pages	6
Datum der Kalibrierung Date of calibration	31.05.2023

Kalibrierscheine ohne Unterschrift haben keine Gültigkeit. *Calibration certificates without signature are not valid.*

Datum Date	Leiter des Kalibrierlaboratoriums Head of the calibration laboratory	Bearbeiter Person in charge
13.06.2023	 Jochen Hohl-Ebinger	 Astrid Semeraro

Die Rückführung der Spektralmessung auf SI-Einheiten erfolgte über den Vergleich mit einer Standardlampe.
The traceability of the measurement of the spectral distribution to SI-Units is achieved using a standard lamp for the calibration of the spectroradiometer.

Identitäts-Nr. / Identity-Nr. :	Kalibrierschein-Nr./ Certificate-Nr. :	Rückführung/ Traceability :
BN-9101-451	40006-20-PTB	PTB

3. Messbedingungen

Measurement conditions

Standardtestbedingungen (STC) / Standard Testing Conditions (STC) :

Absolute Bestrahlungsstärke /
Total irradiance : 1000 W/m²

Nominalwert der Temperatur des
Messobjektes / Nominal Value of
Temperature of the DUT : 25 °C

Spektrale Bestrahlungsstärke /
Spectral irradiance distribution : AM1.5G Ed.4 (2019)

Die Messung der IV-Kennlinie (Strom-Spannungs-Kennlinie) des Messobjektes erfolgt mit Hilfe eines Vierquadranten-Netztes und eines Kalibrierwiderstandes. Die Temperatur der Solarzelle wird mit einem Tastsensor ermittelt und auf (25±0,5)°C eingestellt.

The measurement of the IV-curve is performed with a 4-quadrant power amplifier and a calibration resistor. The temperature of the solar cell is determined by a sensor and adjusted to (25±0.5)°C.

4. Messergebnis

Measurement results

Fläche / Area (da)¹: = (1.0350 ± 0.0064) cm²

¹ : (t) = total area, (ap) = aperture area, (da) = designated illumination area /6/

Kennlinienparameter des Messobjektes unter Standardtestbedingungen (STC) / IV-curve parameter under Standard Testing Conditions (STC) :

	Vorwärtsrichtung / forwards scan direction	Rückwärtsrichtung / reverse scan direction	steady state MPP
V _{OC} =	(1834 ± 18) mV	(1845 ± 19) mV	
I _{SC} (Ed.2 - 2008) =	(20.81 ± 0.40) mA	(20.85 ± 0.40) mA	
I _{MPP} =	18.60 mA	19.03 mA	(19.07 ± 0.58) mA
V _{MPP} =	1502 mV	1560 mV	(1556 ± 35) mV
P _{MPP} =	27.9 mW	29.7 mW	(29.7 ± 1.2) mW
FF =	73.2 %	77.2 %	
η =			(28.7 ± 1.2) %

Angegeben ist jeweils die erweiterte Messunsicherheit, die sich aus der Standardmessunsicherheit durch Multiplikation mit dem Faktor $k=2$ ergibt. Sie wurde gemäß dem "Guide to the expression of Uncertainty in Measurement" ermittelt. Sie entspricht bei einer Normalverteilung der Abweichungen vom Messwert einer Überdeckungswahrscheinlichkeit von 95%.

The expanded measurement uncertainty resulting from the standard measurement uncertainty multiplied with a factor $k=2$ is specified. The calculation was carried out according to the "Guide to the expression of Uncertainty in Measurement". The value corresponds to a Gaussian distribution denoting the deviations of the measurement value within a probability of 95%.

5. Zusatzinformationen

Additional information

Steady State P_{mp}

Supplementary Fig. 32 Certification reports from Fraunhofer ISE for a monolithic perovskite silicon tandem solar cell.

6. According to Figure S1, processing the perovskites in a nitrogen environment is the best condition for IPA. For the rest of the work, the authors investigate different solvents for ambient air. I think throughout the article, N₂ conditions would be a good reference. If this makes a huge difference, there might be a cost-effective solution to integrate the processing in a protected environment. If there is no significant difference, this is already important information for the literature.

Response: We thank the reviewer for the comment. Our experiments demonstrate that perovskite films fabricated in air undergo crystallization more rapidly than those produced in N₂ environment, a phenomenon primarily attributed to moisture's role in accelerating the perovskite formation process, as evidenced by the color change observed in Supplementary Figure 6.

However, it is worth noting that the presence of adsorbed water during the fabrication process can negatively impact the quality and performance of the final perovskite films by promoting excessive PbI₂ formation, shown in Supplementary Figure 8 and Figure 2e.

We now include more detailed experiments and improve the discussion in the revised manuscript.

“Supplementary Figure 6 displays images of perovskite films fabricated using different alcohols, both in N₂ and air environments. These images corroborate the notion that moisture positively affects the crystallization rate of perovskite films⁴⁷, as inferred from the observable color changes.”

Supplementary Fig. 6 Images of films fabricated by various alcohols in different environments. **a, b, c** and **d** in air and **e, f, g** and **h** in N_2 environment.

“Comparatively, perovskite films fabricated in an air environment exhibit a heightened PbI_2 signal (Supplementary Fig. 8 and Fig. 2e), demonstrating that the moisture absorbed during fabrication prompts the decomposition of perovskite films upon air annealing.”

Supplementary Fig. 8 XRD patterns of perovskite films fabricated by different alcohols in N_2 environment.

Fig. 2e XRD patterns of perovskite films fabricated by different alcohols in air environment.

*“For further comparison, we constructed devices under two distinct conditions: an N_2 environment and ambient air, with their respective photovoltaic parameters detailed in **Supplementary Fig. 12**. Devices fabricated in air exhibit smaller V_{oc} compared to those fabricated in N_2 glove box, which can be attributed to moisture-induced film deterioration. More notably, air-fabricated devices generally suffered from pronounced efficiency losses, except for those using nBA solvent. This exception highlights nBA’s resilience to air exposure during fabrication, with such devices achieving the highest conversion efficiency.”*

Supplementary Fig. 12 Photovoltaic parameter of devices fabricated by different alcohols in air and N₂ environment

7. There is an MASCN additive inside the precursor solution, which does not seem like a standard salt. How critical is this? How is the entire process influenced from this (solvent evaporation rate, crystallization control, etc.)?

Response: We appreciate the reviewer's inquiry. The influence of various additives on perovskite film crystallization, including additive-free conditions, MACl, MASCN, and a blend of MACl and MASCN, was thoroughly investigated in our prior research (*Adv. Mater.* 2023, 35, 2207883). Here, we briefly describe the role of MASCN additive on the crystallization of perovskite films in the following.

The introduction of MASCN facilitates the rapid formation of the black α -phase perovskite, attributed to the reduced formation energy provided by the SCN⁻ anion.

During subsequent annealing in ambient air, the moisture absorption promotes grains regrowth, consequently enhancing grain size.

Minor Comments:

1. In the introduction, the authors claim, "Nevertheless, perovskite films with a 1 μm thickness employing...". I wonder why the thickness should be 1 μm ?

Response: Currently, high-efficiency perovskite/silicon tandem solar cells are typically fabricated using a one-step solution method on textured silicon substrates with micrometer-sized pyramids. These textures necessitate perovskite films with thicknesses over 1 μm for full coverage. Increasing the film thickness beyond this threshold, however, leads to diminished solar cell performance due to length limitations of the carrier transport. Carriers generated in the thicker perovskite layers will struggle to reach charge collection layers, adversely impacting tandem cell efficiency. If the thickness of the film is less than 1 μm , the tips of the pyramid may remain exposed, leading to the occurrence of current shunting.

"However, the conventional one-step solution method for depositing perovskite films struggles to achieve full coverage over such pyramid tips due to the limited thickness (less than 1 μm), which is essential for effective carrier transport, thus raising the risk of current shunts^{7,8}."

2. In Figure 1c, can the authors make clear which organic salts were included in the solution?

Response: We labelled the components of the organic salts in the figure1c.

Fig. 1c. Images of organic salts used different alcohols and after exposed in air for 1 h.

3. What is the scan direction in the images in Figure 1d?

Response: The direction we scan is from left to right and has been labelled in the caption of Figure 1.

4. Figure 1 requires more details such as the deposition technique.

Response: We add more details in Fig. 1 with its graphical notes, and more experimental details are added in Methods.

5. Can the authors confirm that all samples in Figure 2 were made in the atmosphere (at what relative humidity) and blade coating?

Response: All of perovskite films and devices were fabricated by blade-coating in air. The air humidity was kept in a range of 35-50 %.

6. Can the authors add the absorptance of the PbI₂ to Figure S5, as they claim PbI₂ causes parasitic absorption?

Response: We tested ultraviolet-visible absorption spectrum for the inorganic

framework and the associated data presentation was added to Supplementary Fig. 12.

“ Additionally, the UV-vis spectra of the inorganic framework is detailed in Supplementary Fig. 9. ”

Supplementary Fig. 9 UV-vis spectra of perovskite films fabricated by various alcohols and inorganic framework.

7. Can the authors show the data points in Figure 3g?

Response: We have shown the data points in Figure 3g.

Fig. 3g. Electroluminescence (EL) spectra for IPA and nBA perovskite device.

8. In Figure 4d, which device's statistical results are reported?

Response: In Figure 4d, we show device's statistical of devices in Figure 4c.

9. Can the authors provide the substrate temperature during blading and the amount of solvent/substrate area, which will increase the precision of their recipe?

Response: We have added a more detailed part of the experiment in the Experiments section. We now improve the discussion in Methods.

“The perovskite was produced using a two-step process. First, PbI₂ and CsBr were co-evaporated by thermal evaporation. The evaporation rate of PbI₂ and CsBr was 4 and 0.8 Å s⁻¹, respectively. The thickness of the inorganic frame is 405nm on a glass substrate and 540 nm on textured silicon substrate. Following, for small area devices

*(2.5 cm * 2.5 cm) 25 μ L organic salt solution was blade-coated with a rate of 20 mm s⁻¹ assisted with N₂ in air, the gap between blade and substrate is 250 μ m and the pressure of the N₂-knife is 30 PSI. As for large area devices (6 cm * 6 cm), the amount of solution was increased to 60 μ L. The ambient temperature is 25 °C and the humidity is 35-50 %. Next, the films were transferred to a hotplate and annealed at 150 °C in air (35% RH) for 20 min.”*

10. Is there any specific reason that the authors use a high evaporation rate of 4 and 0.8 Å s⁻¹ for PbI₂ and CsBr?

Response: Higher evaporation rates are desired for faster fabrication. At those rates, we can obtain a good homogeneity of the inorganic frame on the substrate (M6 size). However, in our equipment, the evaporation rate for PbI₂ is capped at 4 Å s⁻¹ due to limitations of our setup. To ensure the correct perovskite composition, the evaporation rate of CsBr is synchronized with that of PbI₂.

11. A batch of devices showing the comparison of blade/spin coating processes in ambient and N₂ conditions with different solvents would support the claims of the article, specifically the influence of moisture.

Response: On the recommendation of the reviewers, we explored devices fabricated in the N₂ and air environment using different alcohols. We now improve the discussion in the revised manuscript.

*“For further comparison, we constructed devices under two distinct conditions: an N₂ environment and ambient air, with their respective photovoltaic parameters detailed in **Supplementary Fig. 12**. Devices fabricated in air exhibit smaller Voc compared to those fabricated in N₂ glove box, which can be attributed to moisture-induced film deterioration. More notably, air-fabricated devices generally suffered from pronounced efficiency losses, except for those using nBA solvent. This exception*

highlights nBA's resilience to air exposure during fabrication, with such devices achieving the highest conversion efficiency.”

Supplementary Fig. 12 Photovoltaic parameter of devices fabricated by different alcohols in air and N₂ environment.

12. Can the author explain why the overall tandem FF is higher than the bottom cells (Table S6), although the cells are current-matched?

Response: This is due to the fact that our silicon cells have a thinner TCO thickness compared to normal commercial silicon cells. In Si cells, ITO assumes the role of lateral transmission, however, its weak electrical conductivity leads to lower FF. In contrast, the FF in monolithic tandem cells is not constrained by the ITO layer's conductivity, as the device's overall conductivity is not solely dependent on the intermediate ITO.

13. Can the authors give reverse and forward scans together in Figure 4c and f? The

same for Figure 3e.

Response: We present the forward scan data in Figure 4c. For some devices, forward scan data were initially unavailable; thus, we fabricated new devices and updated these data in Figure 4e. Given the comprehensive data already included in Figure 3e, we have relocated the forward scan curves to Supplementary Figure 17.

Supplementary Fig. 17. J - V curves of the champion devices (1.044 cm² aperture area)

Fig. 4c. J - V curves of the tandem device (1.044 cm² aperture area), the digital photo of a device is shown in the inset.

Fig. 4e. J - V curves of the tandem device (16 cm^2 aperture area), the digital photo of a device is shown in the inset.

14. The authors have a strong emphasis on humidity ingress. I wonder what the water content of the solvents is they use. Are they anhydrous? Is this important?

Response: All the reagents we used in our experiments were of anhydrous grade, which is one of the more important parts. Since the water content in the reagents is also a relatively large compared to the amount of moisture absorbed by the films during the fabrication process.

15. It seems 26.3% is not the highest PCE reported for large-area perovskite silicon tandems according to the results in this article (DOI: 10.1126/science.adh3849).

Response: Thanks for your suggestion. We have now revised the manuscript to include more precise expressions.

“Furthermore, we fabricated 36 cm^2 perovskite/silicon tandem cells (aperture area, 16 cm^2), and achieved a conversion efficiency of 26.3% ($V_{oc} = 1.815 \text{ V}$, $J_{sc} = 18.54 \text{ mA cm}^{-2}$, $FF = 78.31\%$), which is among the highest PCE of large-area perovskite/silicon

*tandem cells*¹¹.”

16. Having page and line numbers would be useful to provide feedback.

Response: Thank you very much for your advice. We have added page numbers and line numbers for easier reading.

17. There are several typos in the article, and the authors are recommended to check and correct them.

Response: We have tried our best to correct the typos in the article.

Reviewer #2 (Remarks to the Author):

In the paper “Solvent Engineering for Scalable Fabrication of Perovskite/Silicon Tandem Solar Cells in Air”, the authors report a new solution for wide-bandgap perovskite solar cells and the large-scale practical production of perovskite/silicon tandem solar cells based on industrial pyramid (2-3 μm) silicon heterojunction fabricated in air. They innovatively proposed that the polarity and volatilization rate of solvent together affect the water absorption. This manuscript represents a significant advance in the field of perovskite/silicon tandem solar cells. The conclusions are reasonable and supported by the experimental evidences. I recommend publication after revision:

1. In the paper, authors presented “highly polar alcohols are susceptible to environmental moisture”. The conclusion is unwarranted. Please explain the reasons in details or add some references.

Response: We appreciate your recommendation to elucidate the correlation between solvent polarity and water absorption. In essence, water, being the most polar solvent,

exhibits variable miscibility with alcohols based on their respective polarities, adhering to the 'like dissolves like' principle. This principle dictates that alcohols of higher polarity are more soluble in water, leading to increased water absorption. We have refined our manuscript to reflect these insights more clearly:

“Following the principle that “like dissolves like,”⁴³ the mutual solubility of alcohols and water—and thus their capacity to absorb moisture—is dictated by their polarity difference. Given water's high polarity, alcohols with greater polarity are more soluble in water, leading to increased water absorption.”

2. In Fig. 1c, the color of EA and IPA solutions turned from colorless to light yellow after 1 hour of air exposure. What is the “yellow”? The author should explain the reason clearly to understand the relation between moisture and polarity of alcohol solutions.

Response: In light of reviewer’s suggestion, we will give a more in-depth explanation of the solution discoloration process. In an air environment, the organic salt solution absorbs moisture from the air, leading to rapid oxidation of I^- to I_2 , which cause the solution to turn yellow. This process is accelerated in more polar solvents due to their higher moisture absorption capacity, making them the first to exhibit a yellow colour. We now improve the discussion in the revised manuscript.

*“In an air environment, moisture absorption leads to rapid oxidation of I^- to I_2 , manifesting as a yellowing of the solution^{44,45}. As shown in **Fig. 1c**, the EA and IPA solutions turned from colorless to light yellow after 1 one hour of exposure, while nBA and nPA solutions exhibited no significant color change, underscoring the protective effect of low polarity solvents against moisture interference.”*

3. The author should present more explanations to support the description of “the initial perovskite structure was destructed by residual solution in nPA films, resulting in the formation of white circular stripes”. And, the author should make it clear that

what is “white circular stripes”?

Response: After careful consideration, we think that the residual solution is more inclined to perpetuate the dissolution-recrystallization reaction with the perovskite rather than cause its degradation. To illustrate, we present images of the IPA film captured at different stages of the reaction process, showcasing varying degrees of reaction completion and organic salt excess, as depicted in Supplementary Fig. 4. Further, photographs of the final film states, particularly highlighting the nPA films, reveal significant excess of organic salts, manifesting as a white pattern on the film surface (Supplementary Fig. 5). This occurrence is attributed to incomplete volatilization of the nPA film even after N₂ treatment, allowing the residual solution to continue reacting with the perovskite, leading to the formation of a non-optically active δ -phase and the creation of voids within the material.

Considering that the description of the pattern does not affect our final conclusions, we finally decided to delete this description. We now improve the discussion in the revised manuscript.

*“Fig. 1d illustrates that EA and IPA volatilize fastly and completely after blade-coating, in contrast to nBA and nPA films, which show a gradual darkening. This shift signifies a decrease in volatilization rate with increasing carbon chain length, affecting perovskite crystallization dynamics. However, the slower volatilization rates result in the lingering of residual organic salts, which continue to undergo dissolution-recrystallization reactions with the perovskite⁴⁶. This leads to localized accumulations of organic salts, as evidenced in **Supplementary Fig. 4 and 5.**”*

Supplementary Fig. 4 Images of perovskite films in different reaction stages fabricated by IPA.

a, appropriate amount of organic salt. **b**, slightly excessive organic salts. **c**, excessive organic salts.

Supplementary Fig. 5 Images of final perovskite films fabricated by different alcohols.

4. The author mentioned that “the EA film contained a large amount of PbI_2 , which would remarkably passivate the defects and increased strength of PL signal”. Why does the intensity of PL of EA films in Fig. 2f not increase?

Response: In Fig. 2f, the PL spectra presented are measured with the incident light coming from the glass side. The laser light is absorbed by the unreacted inorganic framework, leading to a diminished PL peak observed in the EA film. To provide a comprehensive view, we also include PL measurements with the incident light directly on the perovskite side, as depicted in Supplementary Fig. 11.

Supplementary Fig. 11 Steady-state PL spectra of perovskite films on glass with the emission from the perovskite film side

5. The PCE of fabricated 36 cm² perovskite/silicon tandem cells (aperture area, 16 cm²), of 26.3% is the highest PCE of large-area perovskite/silicon tandem cells reported. The author should add a table to summarize and make a comparison with other large-area perovskite/silicon tandem cells reported.

Response: We now summarize the recent publications of large-scale perovskite/silicon tandem solar cells (aperture area > 10 cm²) in Supplementary Table 8.

Supplementary Table 8. Comparison of reported large-scale perovskite/silicon tandems

Institution	Fabrication Method	Aperture area	V_{oc} (V)	J_{sc} (mA cm⁻²)	FF (%)	PCE (%)	Ref
UNSW	Spin-coating	16 cm ²	1.74	16.2	78	21.9	[2]
NKU	Hybrid two-step	11.9 cm ²	1.85	17.5	70.6	22.8	[3]
NJU	Hybrid two-step	16 cm ²	1.79	18.5	75.7	25.1	[4]
NKU	Hybrid two-step	11.9 cm ²	1.85	18.1	74.9	25.1	[5]
UNC	Blade-coating	24 cm ²	1.89	18.1	73.6	25.2	[6]
NKU	Hybrid two-step	11.3 cm ²	1.87	18.2	74.9	25.4	[7]
NJU	Hybrid two-step	16 cm ²	1.81	18.5	78.3	26.3	This work

6. In Fig. 4e, the EQE of nBA is higher than IPA between 400-600 nm. The author should explain the reason.

Response: We believe that compared with nBA devices, IPA devices have poor absorption at 400-600 nm, primarily attributed to increased non-radiative recombination at the grain boundaries due to smaller grain sizes. We now improve the discussion in the revised manuscript.

“The integrated J_{sc} value from the external quantum efficiency (EQE) curve in Fig. 3d was calculated to be 20.81 and 20.99 mA cm⁻², respectively, corresponding well with the values obtained from J–V measurements. Compared with the IPA devices, the nBA displayed improved charge collection, particularly between 400 and 600 nm, due to the larger grain sizes minimizing recombination⁴⁹.”

Fig. 3d EQE spectra of the champion device.

7. Why thicker perovskite layers are needed when fabricating on textured silicon even by two-step sequential deposition method? I noticed that the thickness of perovskite is $\sim 1 \mu\text{m}$. How do you determine the thickness of perovskite in tandem solar cells?

Response: In fabricating perovskite films on textured silicon substrates via a hybrid two-step method, we achieved a thickness of around 800 nm. This decision was guided by three critical considerations:

- (i) The need to achieve current matching between the silicon and perovskite subcells necessitates adjusting the perovskite film thickness to match the fixed J_{sc} of silicon.
- (ii) To capture light effectively within the 300-750 nm spectrum, a thicker perovskite layer is essential.
- (iii) For optimal electrical performance, the thickness should be moderated to avoid hindering carrier transport, which could deteriorate the device's electrical performance.

Balancing these factors led us to settle on an optimal perovskite film thickness of approximately 800 nm for our tandem solar cells.

8. The author adjusted the pressure of the N2 knife. And, the according literature, Joule 6, 1–13, August 17, 2022, should be cited.

Response: We now add the recommended references in the revised manuscript.

*“While complete conversion of the inorganic framework to perovskite is achievable through adjustments in parameters like quenching gas pressure and blade-coating rate^{55,56}, such modifications can detract from film uniformity and device performance, as evidenced in **Supplementary Figs. 27-30**. Consequently, parameter tuning was not utilized to fully convert IPA films in tandem devices.”*

[55] Du, M. et al. Surface redox engineering of vacuum-deposited NiOx for top-performance perovskite solar cells and modules. Joule 6, 1931–1943 (2022).

9. The spelling mistakes should be corrected, such as, “siliocn”.

Response: We have tried our best to correct the typos in the article.

Reviewer #3 (Remarks to the Author):

In the manuscript titled with “Solvent Engineering for Scalable Fabrication of Perovskite/Silicon Tandem Solar Cells in Air”, the authors reported the efficiency of 29.4% for perovskite/silicon tandem solar cells and 26.4% for an aperture area of 16 cm². The authors think nBA solvent can mitigate the impact of moisture in air due to the low polarity and moderated volatilization rate. However, how the different polarities of alcohols impact the conversation speed of PbI₂ and the uniformity of perovskite film, the detailed interaction mechanism is unclear. I would recommend to accept it once the authors can address the following comments below.

1. Compared with IPA solvent, nBA solvent can mitigate the impact of moisture in air. The efficiency of nBA device is higher than that of IPA device when the solar cell

devices were fabricated in air. However, the authors should give the photovoltaic performances of nBA and IPA devices when the solar cell devices were fabricated in N₂ environment. Whether nBA solvent can be used in N₂ environment with better performance?

Response: We thanks for the reviewer for the suggestion. We compared the efficiency of the devices s fabricated with different solvents in an N₂ environment and found that devices using nBA did not demonstrate superior efficiency compared to those using IPA; indeed, IPA-based devices showed the highest efficiency in N₂ conditions. However, it's notable that nBA devices fabricated in air outperformed all other devices tested in terms of efficiency. This observation has been incorporated into the revised manuscript to enrich the discussion.

*“For further comparison, we constructed devices under two distinct conditions: an N₂ environment and ambient air, with their respective photovoltaic parameters detailed in **Supplementary Fig. 12**. Devices fabricated in air exhibit smaller V_{oc} compared to those fabricated in N₂ glove box, which can be attributed to moisture-induced film deterioration. More notably, air-fabricated devices generally suffered from pronounced efficiency losses, except for those using nBA solvent. This exception highlights nBA's resilience to air exposure during fabrication, with such devices achieving the highest conversion efficiency.”*

Supplementary Fig. 12 Photovoltaic parameter of devices fabricated by different alcohols in air and N₂ environment.

2. As shown in Figure 1c, why the color of EA and IPA solutions turned to light yellow, the authors should give more detailed explanation about what factors contribute to the change in color and how the reaction occur?

Response: In light of reviewer's suggestion, we will give a more in-depth explanation of the solution discoloration process. In an air environment, the organic salt solution absorbs moisture from the air, leading to rapid oxidation of I⁻ to I₂, which cause the solution to turn yellow. This process is accelerated in more polar solvents due to their higher moisture absorption capacity, making them the first to exhibit a yellow colour. We now improve the discussion in the revised manuscript.

"In an air environment, moisture absorption leads to rapid oxidation of I⁻ to I₂,

manifesting as a yellowing of the solution^{44,45}. As shown in Fig. 1c, the EA and IPA solutions turned from colorless to light yellow after 1 one hour of exposure, while nBA and nPA solutions exhibited no significant color change, underscoring the protective effect of low polarity solvents against moisture interference.”

3. The conversation and crystallinity of perovskite films via two-step method mainly depends on the solvent volatilization rate. How the different polarities of alcohols impact the conversation speed of PbI₂ and the uniformity of perovskite film, the authors should give more detailed explanation.

Response: The effect of the solvent on the reaction is multifaceted, impacting decomposition, conversion, and film homogeneity.

1) **Decomposition of perovskite:** The polarity of the solvent significantly affects its capacity to absorb water, following the principle that 'like dissolves like.' High-polarity solvents, aligning closely with water's high polarity, have enhanced solubility with water and, consequently, tend to absorb more moisture during the fabrication process. This leads to a heightened risk of perovskite decomposition during annealing. Moreover, solvents with low saturated vapor pressure that evaporate slowly are particularly susceptible to water absorption, exacerbating the exposure of PbI₂ to conditions that favor decomposition.

2) **Conversion of perovskite:** The volatilization rate of the solvent is critical. Slower evaporation extends the duration of interaction between the organic salts and the inorganic framework, facilitating a fuller conversion to perovskite.

3) **Homogeneity of the film:** Solvents with lower saturated vapor pressure result in slower evaporation rates, delaying the crystallization process. This deceleration is beneficial for film uniformity, as it allows for a more controlled and even crystallization across the film.

We have refined our discussion in the revised manuscript:

“Following the principle that "like dissolves like,"⁴³, the mutual solubility of alcohols

and water—and thus their capacity to absorb moisture—is dictated by their polarity difference. Given water's high polarity, alcohols with greater polarity are more soluble in water, leading to increased water absorption.”

“Supplementary Fig. 7 reveals a pronounced PbI₂ signal in EA films before annealing, leading to a substantial amount of PbI₂ at the bottom of the perovskite layer (Fig. 2a and 2e). This indicates that the conversion from the inorganic framework to perovskite is incomplete. Such findings suggest that the delay of solvent volatilization rate is conducive to prolonging the reaction of inorganic frameworks with organic salt solutions in terms to promoting the transformation of inorganic framework into perovskite. ”

“We then performed PL mapping test to investigate the homogeneity of the films, as shown in Fig. 2h-j. Given the significant amount of PbI₂ in EA films—which notably passivates defects and enhances the PL signal strength (as detailed in Supplementary Fig. 11)—EA films were excluded from this part of the analysis. The nBA and nPA films demonstrated superior uniformity compared to the IPA film, a trait ascribed to their lower saturated vapor pressure. This characteristic, combined with the solvent's extended chain length, leads to slower volatilization, while reduced polarity further restricts water ingress into the film. Both factors contribute to a diminished crystallization rate of perovskite, yielding films with enhanced homogeneity⁴⁸. However, the slow volatilization rate of the solvent allows the residual solution to continue interacting with the perovskite through dissolution-crystallization reactions. This process tends to produce a non-optically active δ -phase and creates voids within the bulk⁴⁹, culminating in a diminished PL mapping signal in nPA film. ”

4. As shown in Figure 2d, nPA films have been destroyed by residual solution. However, Figure 2f shows the highest PL intensity of nPA films. The enlarged grain size of nPA films can't be clearly observed in SEM image. Moreover, there are large amount of PbI₂ existed on the nPA film surface. The authors should give more related evidence

to prove the increased PL intensity of nPA films.

Response: We thank the reviewer for the comment. Upon reviewing the pertinent data, we observed that the PL signal of the nPA film is weaker than that of both the nBA and IPA films, aligning with our expectations. The confusion arose from unclear color markings in our original figure; hence, we have revised Figure 2f for clarity.

Fig. 2f. PL spectra of perovskite films with the emission from the glass side.

REVIEWER COMMENTS

Reviewer #1 (Remarks to the Author):

I thank to the authors for carefully addressing the queries posed by myself and the other two reviewers. In my previous assessment, I offered various suggestions aimed at refining the precision of the manuscript, primarily consisting of minor adjustments. It appears that the authors have conducted several additional experiments to tackle these inquiries, and with the incorporation of these findings, the manuscript now presents a more robust argument, bolstered by supportive evidence. Furthermore, the authors have similarly attended to the concerns raised by all three reviewers.

Regarding my remaining queries, I am particularly interested in exploring the hysteresis exhibited by these devices, even at the level of individual junctions, following the inclusion of the new J-V curves in the article. Notably, the certified devices report a stabilized power output, which is the preferred method for reporting the PCE. Hence, it raises the question of the statistical results presented throughout the manuscript. Do they solely reflect the reverse scan, the forward scan, or the stabilized PCE? If indeed they pertain to stabilized PCE, I encourage the authors to outline their measurement protocols, including the duration required for stabilization, among other relevant details. Alternatively, if not, it might be beneficial to incorporate forward scans into the statistical distributions, accompanied by an additional overlaying bar chart, distinguished by a different color scheme.

Apart from these considerations, I am of the opinion that the current version of the manuscript is poised for publication.

Reviewer #2 (Remarks to the Author):

My major concerns have been addressed in the revision.

Point-to-point response to the Reviewers' comments

Manuscript #: NCOMMS-24-01196

We sincerely thank all reviewers for their much-valued suggestions, which have enabled us to substantially improve the manuscript's quality. Followings are the detailed actions taken in light of reviewers' comments.

Reviewer #1 (Remarks to the Author):

I thank to the authors for carefully addressing the queries posed by myself and the other two reviewers. In my previous assessment, I offered various suggestions aimed at refining the precision of the manuscript, primarily consisting of minor adjustments. It appears that the authors have conducted several additional experiments to tackle these inquiries, and with the incorporation of these findings, the manuscript now presents a more robust argument, bolstered by supportive evidence. Furthermore, the authors have similarly attended to the concerns raised by all three reviewers.

Regarding my remaining queries, I am particularly interested in exploring the hysteresis exhibited by these devices, even at the level of individual junctions, following the inclusion of the new J-V curves in the article. Notably, the certified devices report a stabilized power output, which is the preferred method for reporting the PCE. Hence, it raises the question of the statistical results presented throughout the manuscript. Do they solely reflect the reverse scan, the forward scan, or the stabilized PCE? If indeed they pertain to stabilized PCE, I encourage the authors to outline their measurement protocols, including the duration required for stabilization, among other relevant details. Alternatively, if not, it might be beneficial to incorporate forward scans into the statistical distributions, accompanied by an additional overlaying bar chart, distinguished by a different color scheme.

Apart from these considerations, I am of the opinion that the current version of the manuscript is poised for publication.

Response: Thanks for your insightful comments. All the statistical results in this article are the initially derived from reverse scanning tests. Following your suggestions, we supplemented more data to include additional statistical distributions.

Our tests on the single junction device with an aperture area of 0.049 cm² revealed no hysteresis, and we found that the efficiencies obtained from reverse scans are consistent with those from stabilized PCE measurements. Consequently, we did not include forward scan results for this configuration in our report, to avoid redundancy. However, we observed hysteresis in both the larger single-junction device with an aperture area of 1.044 cm² and in the perovskite/silicon tandem cells. In response, we conducted further tests on these devices and have now included the forward scan statistical results in the insets of **Figures 3e** and **4d** for a more comprehensive analysis.

Fig. 3e. inset Distributions of 15 devices for each sample is shown inset.

Fig. 4d. inset Distributions of 16 devices for each sample is shown inset.

Reviewer #2 (Remarks to the Author):

My major concerns have been addressed in the revision.

REVIEWERS' COMMENTS

Reviewer #1 (Remarks to the Author):

The authors resolved the minor issues that I raised in the second round of review. I do not have further questions regarding the manuscript.